# Robotic Task Generalization via Hindsight Trajectory Sketches

**Jiayuan Gu[1,2], Sean Kirmani[1], Paul Wohlhart[1], Yao Lu[1], Montserrat Gonzalez Arenas[1],**
**Kanishka Rao[1], Wenhao Yu[1], Chuyuan Fu[1], Keerthana Gopalakrishnan[1], Zhuo Xu[1],**
**Priya Sundaresan[1], Peng Xu[1], Hao Su[2], Karol Hausman[1], Chelsea Finn[1], Quan Vuong[1], Ted Xiao[1]**
[1]Google DeepMind, [2]University of California San Diego

**Abstract:** We propose a policy conditioning method using such rough trajectory sketches, which we call *RT-Trajectory*, that is practical, easy to specify, and allows the policy to effectively perform new tasks that would otherwise be challenging to perform. We find that trajectory sketches strike a balance between being detailed enough to express low-level motion-centric guidance while being coarse enough to allow the learned policy to interpret the trajectory sketch in the context of situational visual observations. In addition, we show how trajectory sketches can provide a useful interface to communicate with robotic policies – they can be specified through simple human inputs like drawings or videos, or through automated methods such as modern image-generating or waypoint-generating methods. We evaluate *RT-Trajectory* at scale on a variety of real-world robotic tasks, and find that *RT-Trajectory* is able to perform a wider range of tasks compared to language-conditioned and goal-conditioned policies, when provided the same training data. Evaluation videos can be found at https://rt-trajectory.github.io/

## 1 Introduction

The pursuit of generalist robot policies has been a perennial challenge in robotics. The goal is to devise policies that not only perform well on known tasks but can also generalize to novel objects, scenes, and motions that are not represented in the training dataset. The generalization aspects of the policies are particularly important because of how impractical and prohibitive it is to compile a robotic dataset covering every conceivable object, scene, and motion. This begs a question: can we design a better conditioning modality that is expressive, practical and, at the same time, leads to better generalization to new tasks?

To this end, we propose to use a *coarse* trajectory as a middle-ground solution between expressiveness and ease of use. Specifically, we introduce the use of a 2D trajectory projected into the camera's field of view, assuming a calibrated camera setup. This approach offers several advantages. For example, given a dataset of demonstrations, we can automatically extract hindsight 2D trajectory labels without the need for manual annotation. In addition, trajectory labels allow us to explicitly reflect similarities between different motions of the robot, which, as we show in the experiments, leads to better utilization of the training dataset resulting in a wider range of tasks compared to language- and goal-conditioned alternatives. Furthermore, humans or modern image-editing models can sketch these trajectories directly onto an image, making it a simple yet expressive policy interface. The main contribution of this paper is a novel policy conditioning framework *RT-Trajectory* that fosters task generalization. This approach employs 2D trajectories as a human-interpretable yet richly expressive conditioning signal for robot policies. Our experimental setup involves a variety of object manipulation tasks with both known and novel objects. Our experiments show that *RT-Trajectory* outperforms existing policy conditioning techniques, particularly in terms of generalization to novel motions, an open challenge in robotics.

## 2 Method

In this section, we describe how we acquire training trajectory conditioning labels from the demonstration dataset. We introduce three basic elements for constructing the trajectory representation format: 2D Trajectories, Color Grading, and Interaction Markers.

7th Conference on Robot Learning (CoRL 2023), Atlanta, USA.

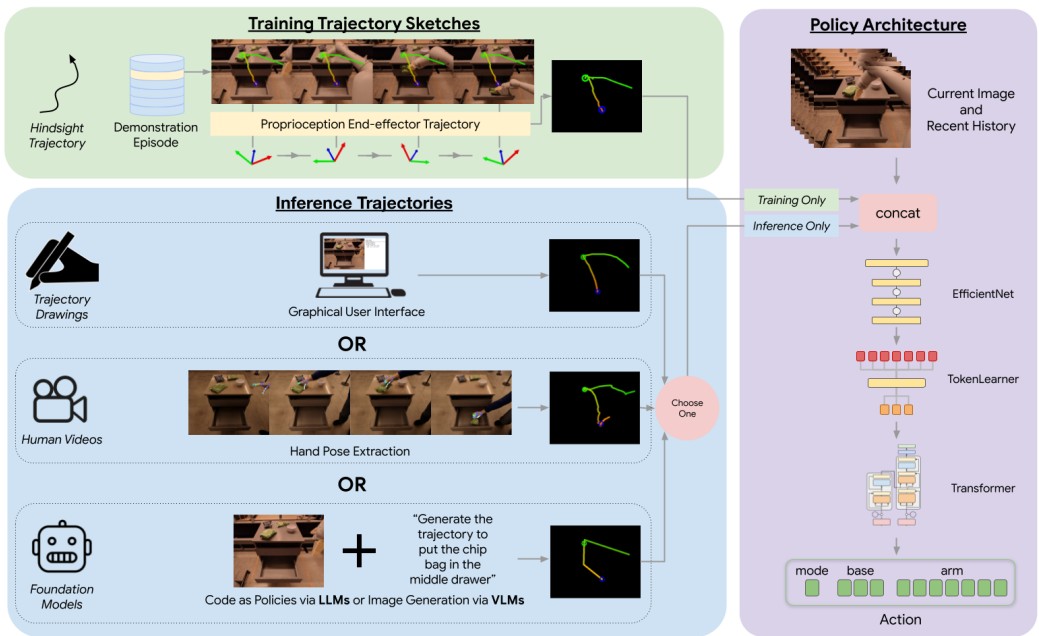

Figure 1: We propose *RT-Trajectory*, a framework for utilizing coarse trajectory sketches for policy conditioning. We train on hindsight trajectory sketches (top left) and evaluate on inference trajectories (bottom left) produced via *Trajectory Drawings*, *Human Videos*, or *Foundation Models*. These trajectory sketches are used as task specification for an RT-1 [1] policy backbone (right).

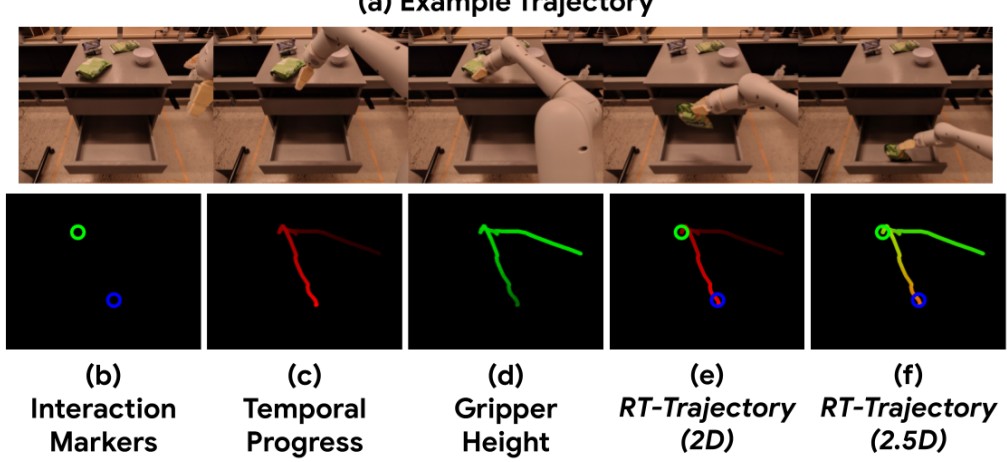

Figure 2: Visualization of the two hindsight trajectory sketch representations we study. Given (a) an example robot trajectory, we extract (b) gripper interaction markers, (c) temporal progress along the 2D end-effector waypoints, and (d) end-effector height. Combining (b) and (c) results in (e) **RT-Trajectory (2D)**, while combining (b), (c), and (d) results in (f) **RT-Trajectory (2.5D)**.

**2D Trajectory** For each episode in the dataset of demonstrations, we extract a 2D trajectory of robot end-effector center points. Concretely, given the proprioceptive information recorded in the episode, we obtain the 3D position of the robot end-effector center defined in the robot base frame at each time step, and project it to the camera space given the known camera extrinsic and intrinsic parameters. We assume that the robot base and camera do not move within the episode, which is common for stationary manipulation. Given a 2D trajectory (a sequence of pixel positions), we draw a curve on a blank image, by connecting 2D robot end-effector center points at adjacent time steps through straight lines.

**Color Grading** To express relative temporal motion, which encodes such as velocity and direction, we also explore using the red channel of the trajectory image to specify the normalized time step $\frac{t+1}{T}$, where

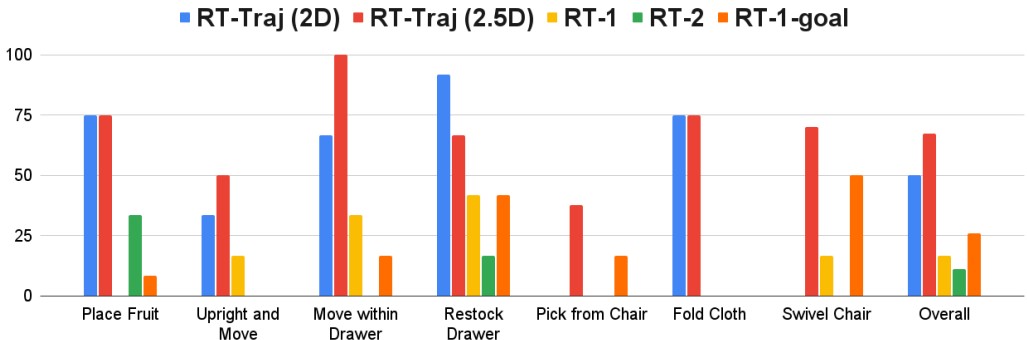

Figure 3: Success rates for unseen tasks when conditioning with human-drawn sketches. Scenarios contain a variety of difficult settings which require combining seen motions in novel ways or generalizing to new motions. Each policy is evaluated for a total of 64 trials across 7 different scenarios.

$T$ is the total episode length. Additionally, we propose incorporating height information into the trajectory representation by utilizing the green channel of the trajectory image to encode normalized height relative to the robot base $\frac{h_{t+1}-h_{min}}{h_{max}-h_{min}}$.

**Interaction Markers** For robot manipulation tasks, time steps when the end-effector interacts with the environment are particularly important. Thus, we explore visual markers that explicitly highlight the time steps when the gripper begins to grasp and release objects. Concretely, we first compute whether the gripper has contact with objects by checking the difference $\delta_t = \hat{p}_t - p_t$ between the sensed ($p_t$) and target ($\hat{p}_t$) gripper joint positions. If the difference $\delta_t > 0$ and $\hat{p}_t > \epsilon$, where $\epsilon$ is a threshold of closing action ($p_t$ increases as the gripper closes), it indicates that the gripper is closing and grasping certain object. If the status change, e.g., $\delta_t < 0 \vee \hat{p}_t \leq \epsilon$ but $\delta_{t+1} > 0 \wedge \hat{p}_{t+1} > \epsilon$, we consider the time step $t$ as a key step for the closing action. Similarly, we can find the key time steps for the opening action. We draw green (or blue) circles at the 2D robot tool center points of all key time steps for closing (or opening) the gripper.

**Trajectory Representations** In this work, we propose two forms of trajectory representation from different combinations of the basic elements. In the first one, *RT-Trajectory (2D)*, we construct an RGB image containing the 2D Trajectory with temporal information and Interaction Markers to indicate particular robot interactions (Fig. 2 (e)). In the second representation, we introduce a more detailed trajectory representation *RT-Trajectory (2.5D)*, which includes the height information in the 2D trajectory (Fig. 2 (f)).

**Policy Training and Inference** We build upon the RT-1 [1] framework and introduce trajectory sketch conditioning. At inference time, we generate trajectory sketches with various input modality sources. These details are fully described in Appendix B.

## 3 Experiments

Our real robot experiments aim to study the following questions:

1. Can *RT-Trajectory* generalize to tasks beyond those contained in the training dataset?
2. Can *RT-Trajectory* trained on hindsight trajectory sketches generalize to different trajectory generation methods at test time, including human-specified or automated methods?
3. Are evaluation rollouts of *RT-Trajectory* meaningfully different from training motions?

We describe our experimental setup in App. C and show what emergent capabilities and realistic case studies that *RT-Trajectory* enable in App. G.

### 3.1 Unseen Task Generalization

In this section, we compare *RT-Trajectory* with other learning-based baselines (RT-1 [1], RT-2 [2], and RT-1-goal) on generalization to unseen task scenarios. The results are shown in Fig. 3 and Table 4. We find that language-conditioned policies struggle to generalize to the new tasks with semantically unseen language instructions, even if motions to achieve these tasks were seen during training (see Sec. F),

compared to *RT-Trajectory*. *RT-Trajectory* (2.5D) outperforms *RT-Trajectory* (2D) on the tasks where height information helps reduce ambiguity. For example, with 2D trajectories only, it is difficult for *RT-Trajectory* (2D) to infer correct picking height, which is critical for `Pick from Chair`.

## 3.2 Automated Trajectory Conditioning

While *RT-Trajectory* is trained on trajectory sketches generated from hindsight end-effector poses from demonstrations and evaluated on human drawing trajectories in Sec. 3.1, we aim to study whether *RT-Trajectory* is able to generalize to trajectories from more automated and general processes at inference time. Specifically, we evaluate quantitatively how *RT-Trajectory* performs when conditioned on coarse trajectory sketches generated by *human video demonstrations*, LLMs via *Prompting with Code as Policies*, and show qualitative results for *image generating VLMs*. Additionally, we compare *RT-Trajectory* against a non-learning baseline (*IK Planner*) to follow the generated trajectories: an inverse-kinematic (IK) solver is applied to convert the end-effector poses to joint positions, which are then executed by the robot.

| Method | Pick | Fold Towel |
| --- | --- | --- |
| IK Planner | 42% | 25% |
| Ours (2D) | 94% | 75% |
| Ours (2.5D) | 100% | 75% |

| Method | Pick | Open Drawer |
| --- | --- | --- |
| IK Planner | 83% | 71% |
| Ours (2D) | 89% | 60% |
| Ours (2.5D) | 89% | 60% |

(a) Trajectory from human video demonstrations.  (b) Trajectory from LLM prompting.

Table 1: Success rate of different trajectory generation approaches across tasks.

**Human Demonstration Videos with Hand-object Interaction** We collect 18 and 4 first-person human demonstration videos with hand-object interaction for `Pick` and `Fold Towel`. An example is shown in Fig. 4. Details about video collection and how we derive trajectory sketches from videos are described in App. D.3. Results are shown in Table 1a.

**Prompting with Code as Policies** We prompt an LLM [3] to write code to generate trajectories given the task instructions and object labels for `Pick` and `Open Drawer`. After executing the code written by the LLM, we get a sequence of target robot waypoints which can then be processed into a trajectory sketch. In contrast with human video trajectories, LLM trajectories are designed to be executed by an IK planner and are therefore precise and linear as seen in Fig. 14. Results are shown in Table 1b.

**Image Generation Models** We condition the VLM with a language instruction and image to output trajectory tokens which are de-tokenized into 2D pixel coordinates for drawing the trajectory on the initial image. Qualitative examples are shown in Fig 5. Although we see that generated trajectory sketches are noisy and quite different from the training hindsight trajectory sketches, we find promising signs that *RT-Trajectory* still performs reasonably. As image-generating VLMs rapidly improve, we expect that their trajectory sketch generating capabilities will improve naturally in the future and be usable by *RT-Trajectory*.

## 3.3 Measuring Motion Generalization

We wish to explicitly measure motion similarity in order to better understand how well *RT-Trajectory* is able to tackle the challenges of novel motion generalization. Towards this, we intend to compare rollout evaluation trajectories to the most similar trajectories seen during training utilizing the discrete Fréchet distance [4] (details in App. F). We find that the proposed novel evaluations indeed require a mix of interpolating seen motions along with generalizing to novel motions altogether, as shown in Fig. 11.

## 4 Conclusion and Limitations

In this work, we propose a novel policy-conditioning for training robot manipulation policies capable of generalizing to tasks and motions that are significantly beyond the training data. Key to our proposed approach is a 2-D trajectory sketch representation for specifying the manipulation tasks. Our trained trajectory sketch-conditioned policy enjoys the controllability from the trajectory sketch guidance, while retaining the flexibility of learning-based policies in handling ambiguous scenes and generalization to novel semantics. We evaluate our proposed approach on 7 diverse manipulation skills that were never seen during training and benchmarked against three baseline methods. Our proposed method achieves a success rate of 67%, significantly outperforming the best prior state-of-the-art methods, which achieved 26%.

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

# A    Related Work

In this section, we discuss prior works studying generalization in robot learning as well as works proposing specific policy conditioning representations.

**Generalization in Robot Learning**  Recent works have studied how learning-based robot policies may generalize robustly to novel situations beyond the exact data seen during training. Empirical studies have analyzed generalization challenges in robotic imitation learning, focusing on 2D control [5], demonstration quality [6], visual distribution shifts [7], and action consistency [8]. In addition, prior works have proposed evaluation protocols explicitly testing policy generalization; these include generalizing to novel semantic attributes [9], holdout language templates [10], unseen object categories [11, 12, 13, 14], new backgrounds and distractors [15, 16], combinations of distribution shifts [1, 17], open-set language instructions [18, 19], and web-scale semantic concepts [2]. While these prior works largely address semantic and visual generalization, we additionally study task generalization which include situations which require combining seen states and actions in new ways, or generalizing to wholly unseen states or motions altogether.

**Policy Conditioning Representations**  We examine a few approaches for policy conditioning. Broadly, there are 2 axes to consider: (1) over-specification and under-specification of goals, and (2) conditioning on all states in a trajectory versus only the end state. The most prolific recent body of work focuses on language-conditioned policies [10, 1, 2, 20, 21, 22, 23], which utilize templated or freeform language as task specification. Language-conditioned policies can be thought of as *under-specified on the end state* (e.g. there are many possible end-states for a policy that completes `pick can`). There are many image-conditioned policy representations with the most popular technique being goal-image conditioning: where a final goal image defines the desired task's end-state [24, 25]. Goal image conditioned policies can be thought of as *over-specified on the end state* (i.e. "what to do") because they define an entire configuration, some of which might not be relevant. For example, the background pixels of the goal image might not be pertinent to the task, and instead contain superfluous information. There are some examples of intermediate levels of specification that propose 2D and 3D object-centric representations [14, 9, 19], using a multimodal embedding that represents the task as a joint space of task-conditioned text and goal-conditioned image [18, 17, 9], and describing the policy as code [26] which constrains how to execute every state. An even more detailed type of state-specification would be conditioning on an entire RGB video which is equivalent to *over-specification over the entire trajectory of states* (i.e. "how to do it") [27]. However, encoding long videos in-context is challenging to scale, and learning from high-dimensional videos is a challenging learning problem [10]. In contrast, our approach uses a lightweight coarse level of state-specification, which aims to strike a balance between sufficient state-specification capacity to capture salient state properties while still being tractable to learn from. We specifically compare against language-conditioning and goal-image conditioning baselines, and show the benefits of using a mid-level conditioning representation such as coarse trajectory sketches.

# B    Implementation Details

## B.1    Policy Training

We leverage Imitation Learning due to it's strong success in multitask robotic imitation learning settings [10, 24]. More specifically, we assume access to a collection of successful robot demonstration episodes. Each episode $\tau$ contains a sequence of pairs of observations $o_t$ and actions $a_t$: $\tau = \{(o_t, a_t)\}$. The observations include RGB images obtained from the head camera $x_t$ and hindsight trajectory sketch $c_{traj}$. We then learn a policy $\pi$ represented by a Transformer [28] using Behavior Cloning [29] following the RT-1 framework [1], by minimizing the log-likelihood of predicted actions $a_t$ given the input image and trajectory sketch. To support trajectory conditioning, we modify the RT-1 architecture as follows. The trajectory sketch is concatenated with each RGB image along the feature dimension in the input sequence (a history of 6 images), which is processed by the image tokenizer (an ImageNet pretrained EfficientNet-B3). For the additional input channels to the image tokenizer, we initialize the new weights in the first convolution layer with all zeros. Since the language instruction is not used, we remove the FiLM layers used in the original RT-1.

## B.2    Trajectory Conditioning during Inference

During inference, a trajectory sketch is required to condition *RT-Trajectory*. We study 4 different methods to generate trajectory sketches: *human drawings*, *human videos*, *prompting LLMs with Code as Policies*, and *image generation models*.

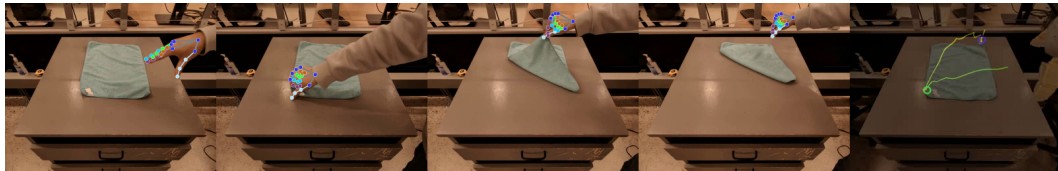

Figure 4: Trajectory from human demonstration video to fold a towel. From left to right, the first 4 images show the human demonstration, and the last image shows the derived trajectory sketch.

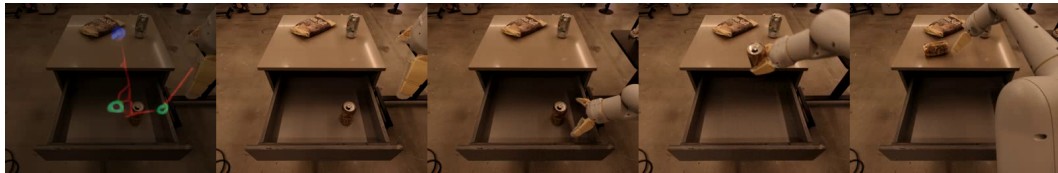

Figure 5: Example trajectory from image generation models. From left to right, the first image shows the overlaid trajectory sketch, and the next 4 images show the rollout conditioned on it. The language instruction is: `pick orange can from top drawer and place on counter`.

**Human-drawn Trajectory Sketches** Human-drawn sketches are an intuitive and practical way to generate trajectory sketches at inference time. To scalably produce these sketches during evaluations, we design a simple UI for users to draw trajectory sketches given the robot's initial camera image, as shown in App. D.1.

**Human Demonstration Videos with Object Interaction** We also study first-person human single-hand demonstration videos. We estimate the trajectory of human hand poses from the video, and convert it to a trajectory of robot tool poses, which can be used to generate a trajectory sketch.

**Prompting LLMs with Code as Policies** Large Language Models have demonstrated the ability to write code to perform robotics tasks [26]. In this work, we aim to leverage the spatial and physical reasoning capabilities of LLMs to output trajectories. We follow a similar recipe as described in [30] to build a prompt which contains text descriptions about the objects in the scene detected by a VLM, the robot constraints, the gripper orientations and coordinate systems, as well as the task instruction. By using this prompt, the LLM writes code to generate a series of 3D poses - originally intended to be executed with a motion planner, which we can then re-purpose to draw the trajectory sketch on the initial image to condition *RT-Trajectory*.

**Image Generation Models** Since our trajectory conditioning is represented as an image, we can use text-guided image generation models ([31, 32, 33, 34]) to generate a trajectory sketch provided the initial image and language instruction which describes the task. In our work, we use a PaLM-E style ([35]) model that generates vector-quantized tokens derived from ViT-VQGAN ([36]) that represent the trajectory image. Once detokenized, the resulting image can be used to condition *RT-Trajectory*.

## C   Experiment Details

### C.1   Experimental Setup

We use a mobile manipulator robot from Everyday Robots in our experiments, which has a 7 degree-of-freedom arm, a two-fingered gripper, and a mobile base.

**Seen Skills** We use the RT-1 [1] demonstration dataset for training, which contains 542 instructions inspired by an office kitchen setting. The language instructions consist of 8 different manipulation skills (e.g., `Move Near`) operating on a set of 17 household kitchen items; in total, the dataset consists of 73,334 real robot demonstrations across these 542 seen tasks, which were collected by manual teleoperation. A more detailed overview is shown in Table 2.

**Unseen Skills** We propose 7 new evaluation skills which include unseen objects and manipulation workspaces, as shown in Table 3 and Fig. 3. `Place Fruit` inspects whether the policy can place objects into unseen containers. Both `Upright and Move` and `Move within Drawer` examine whether the policy can combine different seen skills to form a new one. For example, `Move within Drawer` studies whether the policy is able to move objects within the drawer while the seen skill `Move Near` only covers those motions at height of the tabletop. `Restock Drawer` requires the robot to place snacks into the drawer at

| Skill | Count | Description | Example Instruction |
|---|---|---|---|
| Pick Object | 17 | Lift the object off the surface | pick coke can |
| Move Object Near Object | 337 | Move the first object near the second | move pepsi can near rxbar blueberry |
| Place Object Upright | 8 | Place an elongated object upright | place water bottle upright |
| Knock Object Over | 8 | Knock an elongated object over | knock redbull can over |
| Open Drawer | 3 | Open any of the cabinet drawers | open the top drawer |
| Close Drawer | 3 | Close any of the cabinet drawers | close the middle drawer |
| Place Object into Receptacle | 84 | Place an object into a receptacle | place brown chip bag into white bowl |
| Pick Object from Receptacle and Place on the Counter | 82 | Pick an object up from a location and then place it on the counter | pick green jalapeno chip bag from paper bowl and place on counter |
| Total | 542 | | |

Table 2: The list of seen training tasks with their descriptions and example language instructions. Language instructions are only used for language-conditioned baselines.

an empty slot. It studies whether the policy is able to place objects at target positions precisely. `Pick from Chair` investigates whether the policy can pick objects at an unseen height in an unseen manipulation workspace. `Fold Towel` and `Swivel Chair` showcase the capability to manipulate a deformable object and interact with an underactuated system.

| Skill | Counts | Description | Example instruction |
|---|---|---|---|
| Place Fruit | 12 | Place fruit into the container | place orange into basket |
| Upright and Move | 6 | Place an object upright *and* move it near another | place green can upright near pepsi can |
| Move within Drawer | 6 | Move one object near another *within* the drawer | move coke can near 7up can at top drawer |
| Restock Drawer | 12 | Place objects into the desired position in the drawer | place coke can into the top right of top drawer |
| Pick from Chair | 8 | Pick an object placed on the chair | pick apple from chair |
| Fold Towel | 4 | Fold the towel by moving one corner to another | fold towel from bottom right |
| Swivel Chair | 10 | Swivel the office chair | push the chair |

Table 3: The list of unseen evaluation tasks with their descriptions and example language instructions. Language instructions are only used for language-conditioned baselines. "Counts" refers to the number of scenes collected for evaluation.

**Evaluation Protocol** Different trajectory sketches will prompt *RT-Trajectory* to behave differently. To make the quantitative comparison between different methods as fair as possible, we propose the following evaluation protocol. For each skill to evaluate, we collect a set of *scenes*. Each scene defines the initial state of the task, described by an RGB image taken by the robot head camera. During evaluation, we first align relevant objects to the initial state specified by the *scene*'s RGB image, and then run the policy. For conditioning *RT-Trajectory*, we use human-drawn sketches for the unseen tasks in Sec. 3.1. Additionally, in Sec. 3.2, we evaluate the automated trajectory sketch generation approaches presented in Sec. B.2.

### C.2 Implementation Details for *RT-1-goal*

The network architecture of *RT-1-goal* is the same as *RT-Trajectory*, except a goal image is used instead of a trajectory sketch. To acquire goal conditioning for training, we use the last observation of each episode as the goal image for all frames in the episode. For the goal images used for policy conditioning in the quantitative evaluation in Sec. 3.1, we pre-save goal images similar to how we pre-save the initial scene configuration or trajectory sketch goals.

### C.3 Quantitative Results for Unseen Tasks

See Table 4.

## D Implementation Details for Different Input Modalities

### D.1 Human-drawn Trajectory Sketches

As the main trajectory generation method we study is user-specified trajectory drawings, we develop a graphical user interface (GUI) for users to draw trajectory sketches. See Fig. 6 for example. Given the current robot camera image, a user can drag and move the mouse to draw curves on the canvas. Then,

| Task | RT-Traj (2D) | RT-Traj (2.5D) | RT-1 | RT-2 | RT-1-goal |
|------|--------------|----------------|------|------|-----------|
| Place Fruit | 75 | 75 | 0 | 33.3 | 8.3 |
| Upright and Move | 33.3 | 50 | 16.7 | 0 | 0 |
| Move within Drawer | 66.7 | 100 | 33.3 | 0 | 16.7 |
| Restock Drawer | 91.7 | 66.7 | 41.7 | 16.7 | 41.7 |
| Pick from Chair | 0 | 37.5 | 0 | 0 | 16.7 |
| Fold Towel | 75 | 75 | 0 | 0 | 0 |
| Swivel Chair | 0 | 70 | 16.7 | 0 | 50 |
| Overall | 50 | 67.2 | 16.7 | 11.1 | 26 |

Table 4: Success rates for unseen tasks when conditioning with human-drawn sketches.

they can click on the canvas to add markers to indicate gripper closing or opening actions. Additionally, the UI interface also supports simple height annotation. Users can specify the desired height values for pixels they select on the canvas. This height value will be assigned to the closest point on the drawn 2D trajectory. For unannotated points on the 2D trajectory, we interpolate their height values according to annotated ones.

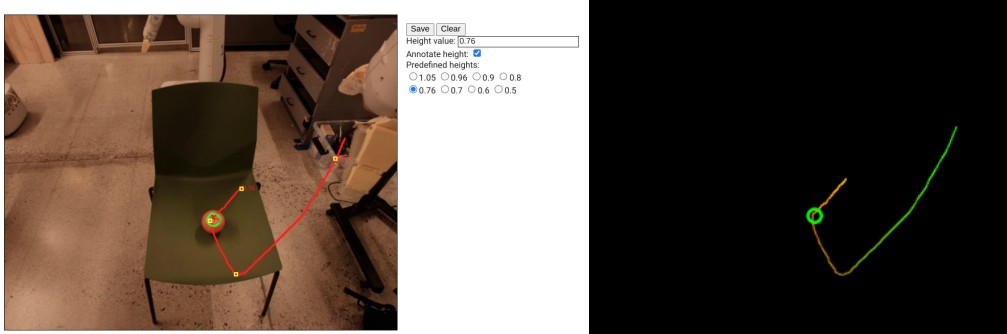

Figure 6: Left: The GUI for users to draw trajectory sketches given the robot's current camera image. The 2D trajectory is directly drawn by manual input, which can then be annotated with interaction markers or waypoints corresponding to user-specified heights. Right: The resulting height-aware trajectory sketch generated according to the output of the UI.

### D.2 Generating Human-drawn Trajectory Sketches

For each scene, we use a held-out *RT-Trajectory* (2.5D) policy to explore different trajectory "prompts" given a budget of (10) trials, and save the trajectory sketch of the first successful episode. We refer to such process as "prompt engineering" (Sec. G). If all attempts fail, we just save the trajectory sketch from the last episode. *RT-Trajectory* policies used for evaluation are trained with different random seeds and evaluated with the saved trajectory sketches as conditioning. Note that we observe that our evaluated policies can have non-zero success rates on the scenes where we fail to find a successful episode during "prompt engineering". For *RT-1-goal*, the image of the last step of the episode used to generate the trajectory sketch for each scene is saved as the goal conditioning for evaluation.

### D.3 Human hand pose estimation

We employ Mediapipe [37] to detect the human hand pose represented as 21 landmarks from the 2D image at each video frame. The two landmarks on the thumb and another two landmarks on the index finger are used to represent a parallel gripper. The 2D landmarks are lifted to 3D given the depth map. We then interpolate the tool pose from these four points. We manually annotate the key frames at which the hand begins to grasp and release the target object. Given estimated tool poses and key frames for interaction, we can generate a trajectory sketch per video.

# E Motion Diversity Analysis

## E.1 Computation of trajectory similarity

To measure the distance between two end-effector motion trajectories we employ the Fréchet distance [4, 38], a measure that quantifies the similarity between two curves by finding the minimum "leash length" required for two agents traversing each curve simultaneously while maintaining their respective temporal order. As a well-adopted similarity measure in computer vision and vehicle tracking [39], Fréchet distance may be a reasonable choice for comparing 3D robot end-effector waypoint trajectories since it is order-preserving and parameterization independent [40].

Specifically, consider two trajectories $\tau$ and $\tau'$ where each trajectory contains $n$ pose waypoints consisting of the sensed end-effector center positions, $\tau = \{\rho_0, \rho_1, ..., \rho_m\}$ and $\tau' = \{\rho'_0, \rho'_1, ..., \rho'_n\}$, and $d(\rho_i, \rho'_i)$ is a distance measure like Euclidean distance. Then, using the notation $\tau[1:]$ to denote removing the first element and returning the rest of the sequence $\tau$, the Fréchet distance $F_D$ is recursively defined as:

$$F_D(\tau, \tau') = \max(d(\rho_0, \rho'_0), \min\{F_D(\tau[1:], \tau'[1:]), F_D(\tau, \tau'[1:]), F_D(\tau[1:], \tau')\})$$

## E.2 Additional samples of trajectory similarities

Figure 7 shows additional examples of evaluation trajectories and their most similar trajectories in the training dataset.

# F Measuring Motion Generalization

We wish to explicitly measure motion similarity in order to better understand how well *RT-Trajectory* is able to tackle the challenges of novel motion generalization. Towards this, we intend to compare rollout evaluation trajectories to the most similar trajectories seen during training. To accomplish this, we propose to measure trajectory similarity by utilizing the discrete Fréchet distance [4] (details in App. E.1). By computing the distance between a query trajectory and all trajectories in our training dataset we can retrieve the most similar trajectories our policy has been trained on. We perform this lookup for trajectories from the rollouts for the novel motion evaluations in Sec. 3.1 and present the results in Fig. 11. We find that the trajectories for novel tasks have varying amounts of similar trajectories in the training trajectories. In some cases, e.g., the motion for `place a fruit in a tall bowl` may be very similar to the motion for an already seen instance of `move X near Y`. However, for many novel skills the most similar examples in the training data are still significantly more different than for examples within the training set, as can be seen in the distribution of their Fréchet distances. In addition, even for evaluation trajectories that seem close in shape to the most similar training trajectories, we find differences in precision-critical factors like the z-height of gripper interactions (picks that are just a few centimeter incorrect will not succeed) or semantic relevance (the most similar training trajectories describe different skills than the target trajectory). Thus, we expect that the proposed novel evaluations indeed require a mix of interpolating seen motions along with generalizing to novel motions altogether.

# G Emergent capabilities and case studies

**Prompt Engineering for Robot Policies** Similar to how LLMs respond differently in response to language prompt engineering, *RT-Trajectory* enables *visual* prompt engineering, where a trajectory-conditioned policy may exhibit better performance when the initial scene is fixed but the coarse trajectory prompts are improved. We find that changing trajectory sketches induces *RT-Trajectory* to change behavior modes in a reproducible manner, which suggests an intriguing opportunity: if a trajectory-conditioned robot policy fails in some scenario, a practitioner may just need to "query the robot" with a different trajectory prompt, as opposed to re-training the policy or collecting more data. Qualitatively, this is quite different from standard development practices with language-conditioned robot policies, and may be viewed as an early exploration into zero-shot instruction tuning for robotic manipulation, similar to capabilities seen in language modeling [41].

**Generalizing to Realistic Settings** Prior works studying robotic generalization often evaluate only a few distribution shifts at once, since generalizing to simultaneous physical and visual variations is challenging; however, these types of simultaneous distribution shifts are widely prevalent in real world settings. As a qualitative case study, we evaluate *RT-Trajectory* in 2 new buildings in 4 realistic novel rooms which

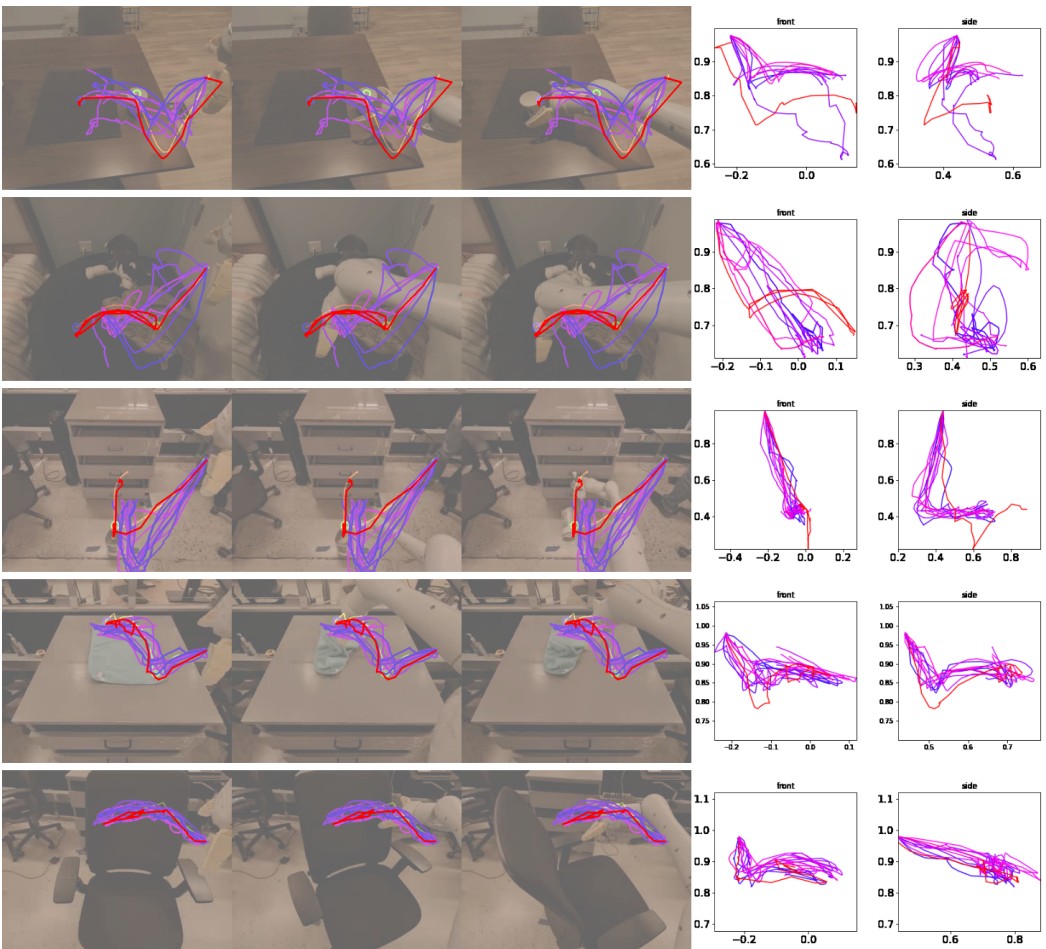

Figure 7: Evaluation trajectories for new skills and their 10 closest trajectories from the training set. Each row shows three frames of a skill evaluation rollout, with the executed trajectory and similar training set trajectories overlaid, as well as depicting the trajectories in an orthographic projection in robot base frame looking at the robot from the front and the side. As can be seen, the policy is able to follow the desired trajectories closely and achieve the tasks. While in many cases, in particular in image space, some of the similar trajectories from the training set look very close to the executed trajectory, the front and side view in rows 1 to 4 reveal that the policy at some crucial point has to - and successfully does - deviate from what it has seen during training. E.g., in row 3 the prompt says to go all the way down to pick up a bottle, while all nearest training trajectories are from `close middle drawer`, which doesn't move the gripper down far enough. Additionally, row 5 is an example where for a `swivel chair` prompt trajectory there coincidentally are many very closely matching `move X near Y` training trajectories. However, the prompt here specifies to not close the gripper at the first contact point, which the policy is able to respect.

contain entirely new backgrounds, lighting conditions, objects, layouts, and furniture geometries. With little to moderate trajectory prompt engineering, we find that *RT-Trajectory* is able to successfully perform a variety of tasks requiring novel motion generalization and robustness to out-of-distribution visual distribution shifts. These tasks are visualized in Fig. 12 and rollouts are shown fully in Fig. 13.

**Retry behavior** Compared to non-learning methods, *RT-Trajectory* is able to recover from execution failures. Fig. 14 illustrates the retry behavior emerged when *RT-Trajectory* is opening the drawer given the trajectory sketch generated by the CaP (Sec. B.2). After a failure attempt to open the drawer by its handle, the robot retried to grasp the edge of the drawer, and managed to pull the drawer.

**Height-aware Disambiguation for *RT-Trajectory* (2.5D)** 2D trajectories (without depth information) are visually ambiguous for distinguishing whether the robot should move its arm to a deeper or higher. We

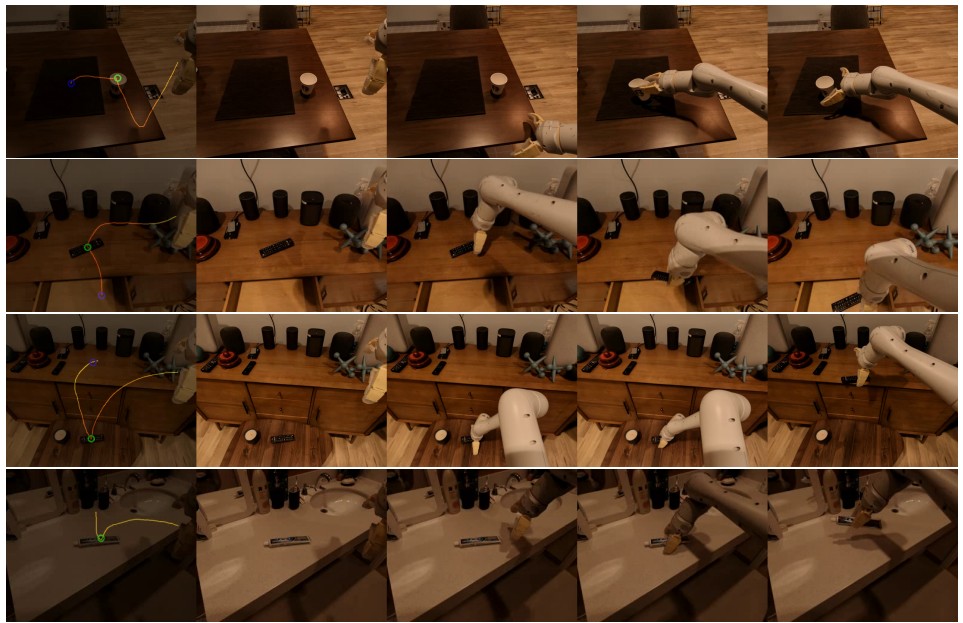

Figure 8: Visualizing additional interesting examples of *RT-Trajectory*'s generalization performance in new scenarios. These include a novel kitchen room setting with an unseen cup and unseen placemat, a new living room room setting with new manipulation objects with new furniture pieces in new heights, and a bathroom setting with harsh lighting and different table height.

find that height-aware color grading for *RT-Trajectory* (2.5D) can effectively help reduce such ambiguity, as shown in Fig. 15.

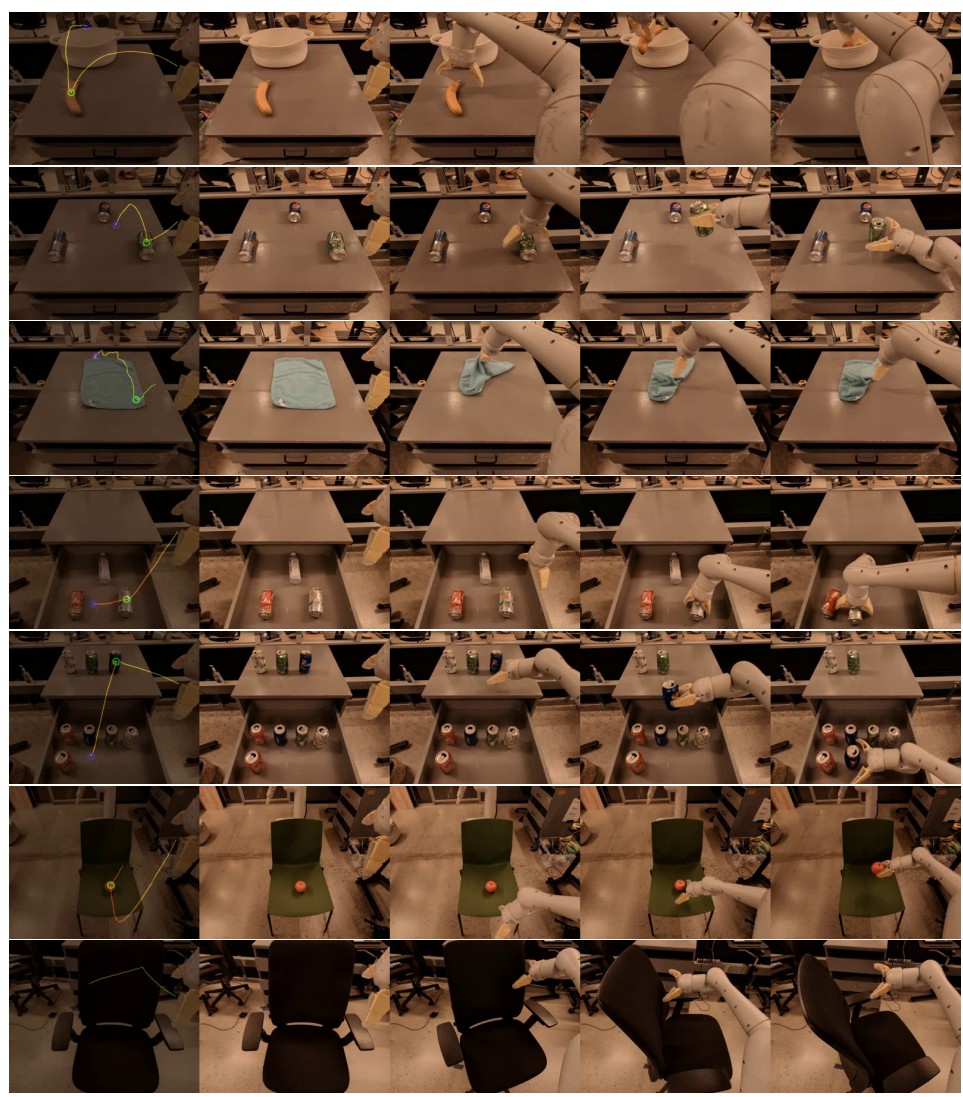

Figure 9: Example rolllouts of 7 unseen skills. The trajectory sketch overlaid on the initial image is visualized. From top to bottom: `Place Fruit`, `Upright and Move`, `Fold Towel`, `Move within Drawer`, `Restock Drawer`, `Pick from Chair`, `Swivel Chair`.

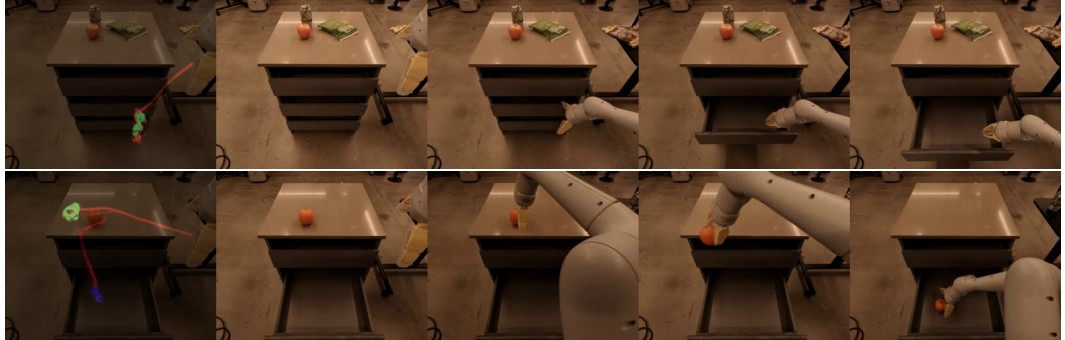

Figure 10: Example trajectories from image generation models. Each row shows the trajectory sketch overlaid on the first frame and the rollout. The language instructions are: `open middle drawer` and `place orange into middle drawer`.

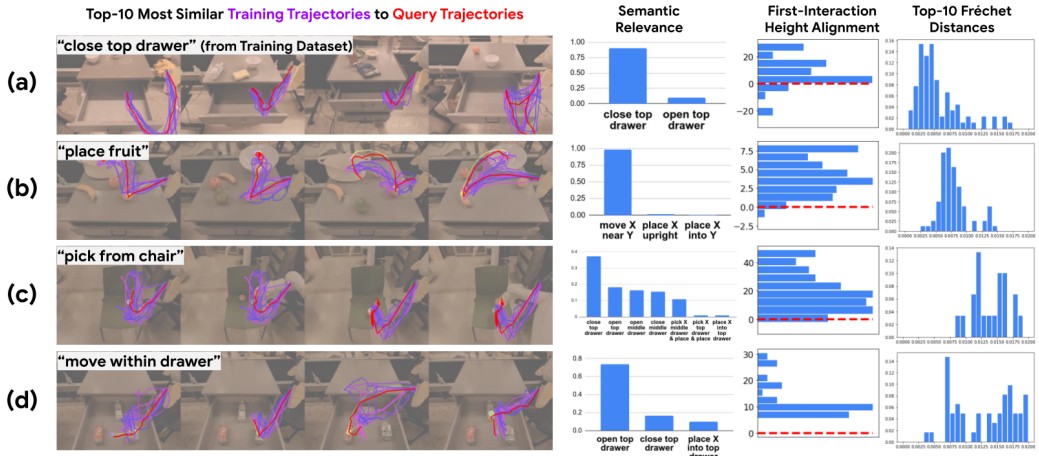

Figure 11: Similarity of motion trajectories to examples in the training set. Each visualization on the left shows an initial image of a rollout and super-imposed the executed trajectory (red) as well as the 10 most similar trajectories in the database (purple). Row (a) shows examples of the in-distribution `close top drawer` skill seen in the training data. Rows (b,c,d) show novel evaluation rollouts. The charts on the right show statistics of the most similar training samples over a larger set of queries for the skill of the row, such as the distribution of skill semantics of the samples, the height of the end-effector at the point of the first interaction with the environment, and the distribution of Fréchet distances. While some query trajectories are well represented in the training data, for many the closest matches are quite different, come from unrelated semantic categories and from the required trajectory at crucial interaction points.

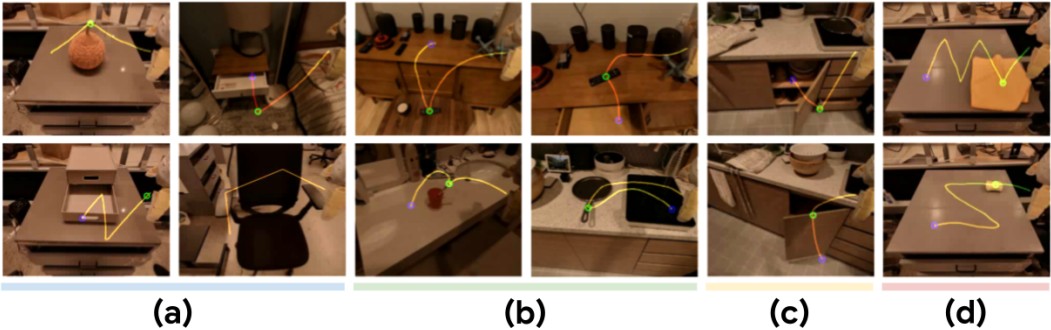

Figure 12: Example *RT-Trajectory* evaluations in realistic scenarios involving (a) novel articulated objects requiring new motions, (b) manipulation on new surfaces in new buildings in new heights, (c) interacting with a pivot-hinge cabinet despite training only on sliding-hinge drawers, and (d) circuitous tabletop patterns extending beyond direct paths in the training dataset. Full rollouts are shown in the supplemental video at https://rt-trajectory-anon.github.io/

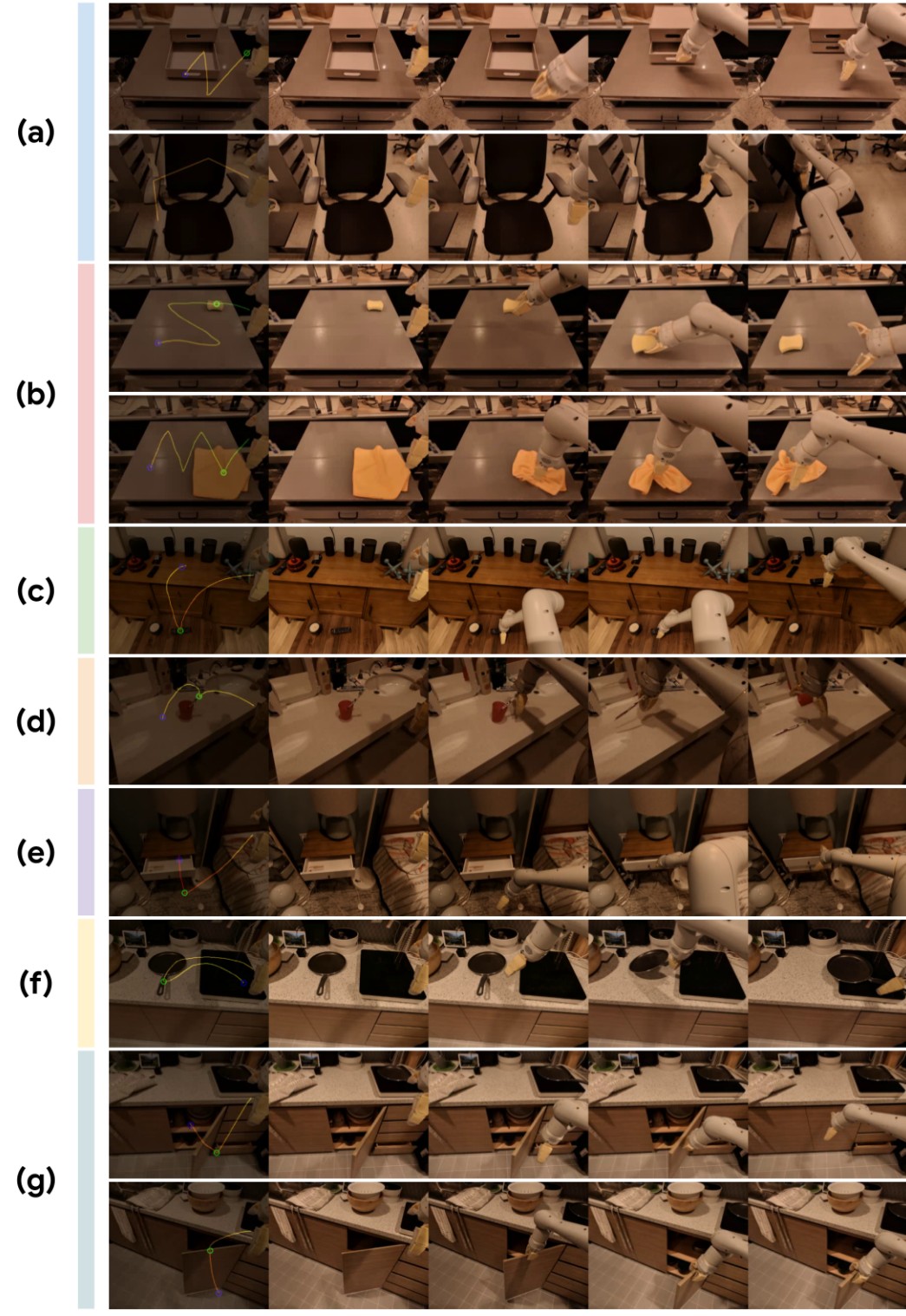

Figure 13: Qualitative examples of emergent capabilities of *RT-Trajectory* in realistic scenarios beyond the training settings: (a) new articulated objects requiring novel motion strategies, (b) new circuitous motions requiring multiple turns, (c) new living room setting with a new height, object, and background, (d) new bathroom setting with precise picking from a cup, (e) new bedroom setting with a drawer at a new height, (f) new kitchen setting with a new pivot hinge that requires a new motion for opening and closing, and (g) new kitchen setting with an unseen pan requiring placement onto a new stove.

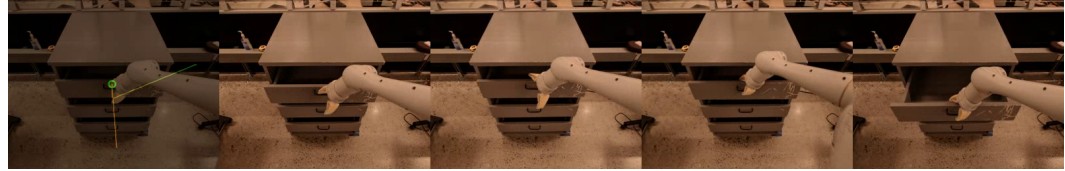

Figure 14: Example of retry behavior. The first image is the trajectory sketch generated from the CaP overlaid on the initial observation. The remaining images show the rollout. The robot first attempts to open the drawer by grasping its handle, but fails (2nd image). Then, it retries to open the drawer by grasping the edge instead.

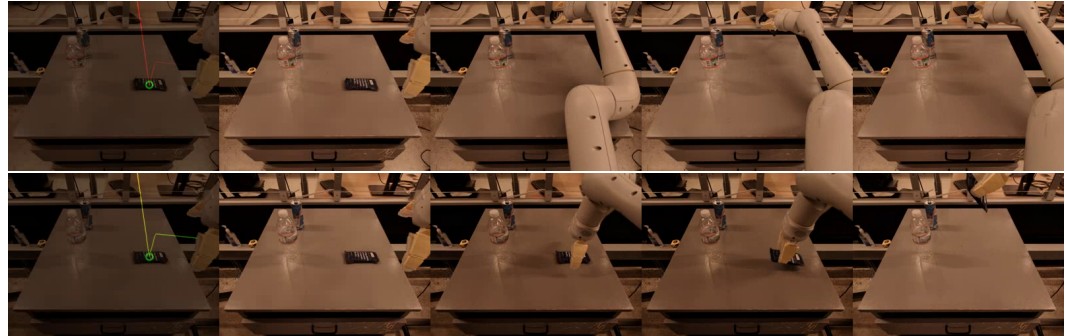

Figure 15: Comparison between *RT-Trajectory* (2D) and *RT-Trajectory* (2.5D). Given the same 2D trajectory generated by the CaP, *RT-Trajectory* (2.5D) lifts the object while *RT-Trajectory* (2D) moves the object to a deeper position due to the ambiguity of a 2D trajectory.

