# OpenReview forum: "Robotic Task Generalization via Hindsight Trajectory Sketches"
_robot-learning.org/CoRL/2023/Workshop/OOD — OOD Workshop @ CoRL 2023_

### Official Review · Reviewer_AQZ9 · 2023-10-13
**Improved Policy Conditioning to Induce Generalization**

**Rating:** 8
**Confidence:** 4

**Review:**

The paper proposes a policy conditioning method, RT-Trajectory, that utilizes physical motion and actuation ('trajectories') in lieu of language or goal specification to promote generalization in learning from demonstrations. The procedure encapsulates the intuition that physical trajectories form a clearer `distance metric' between tasks than image or language grounding alone, while being more natural to construct in an automated fashion. The method is validated in a variety of hardware and task setups and demonstrates both improved performance with respect to baselines on task completion and nascent out-of-distribution generalization for new tasks that require similar behavior despite semantic or visual differences from training data.

The paper fits naturally into the empirical side of OOD generalization, in which (a la Fig. 11) effective policies for addressing new tasks can be synthesized from an effective conditioning `embedding.' Additionally, there is the added benefit of improved robustness for within-task generalization (e.g., to new objects). The writing clearly presents the work, which builds on recent state-of-the-art developments building on the success of VLMs and LLMs. The hardware results are extensive and compare to a strong suite of baseline policies.

One point of feedback is the clarification of how the conditioning would operate 'in situ.' That is, for 'under-specified' methods (e.g., language), the implementation in the real world seems direct: the robot gets a language command and executes the policy. For 'over-specified' methods (e.g., goal image conditioning), the practical implementation may be more challenging (how to generate the goal image in practice?). While RT-Trajectory seems to (intentionally) alleviate some of the latter concern, the impediments to practical implementation still seem non-trivial. Additional clarification of what subset of the four conditioning methods are best suited to 'real-time' or 'real-world' operation would be valuable.

---

### Official Review · Reviewer_LXWA · 2023-10-16
**Policy conditioning on trajectory from various input modules with strong hardware demonstration**

**Rating:** 9
**Confidence:** 4

**Review:**

- Summary:
    - The authors propose trajectory-conditioned policy learning and demonstrate generalization to previously unseen tasks and environments. The authors describe trajectory representation, automated trajectory conditioning with various input modalities, and demonstrate high success rates on a variety of hardware experiments.
The submission is relevant to the workshop topic, well-organized and clearly written.
The paper is novel in proposing a midground conditioning method balancing expressiveness and ease of implementation.
The research question being pursued is a better conditioning modality in terms of expressiveness, practicality, and generalization abilities, which is a significant problem.
- Strengths
    - The proposed method is verified extensively on hardware.
    - Generating trajectory representations that encode temporal and height information seems to be useful beyond this particular work.
    - Inference conditioning can be hand-drawn or automated, and three automation modalities are tested to show success for various tasks.
- Weaknesses
    - Training data on previously successful robot demonstrations seems nontrivial to acquire, but the proposed method can serve as an improvement of previous policy, especially on generalization.
    - Does the camera need to be recalibrated every time the robot moves to a new location?
    - Line 320: figure reference missing

---

### Decision · Program_Chairs · 2023-10-17

**Decision:**

Accept

**Comment:**

We agree with the reviewers’ assessment that this work is technically sound and will contribute to productive, topical discussions at the 2023 Workshop on OOD Generalization in Robotics. In particular, we appreciate that the authors' hypothesis on effective representations for generalization is verified extensively on hardware. We recommend the authors incorporate the reviewers’ feedback into their camera-ready submission to further improve their manuscript.